# Arbitrary-Shaped Image Generation via Spherical Neural Field Diffusion

**Jiyuan Xia, Yuanshen Guan, Ruikang Xu, Zhiwei Xiong** *

University of Science and Technology of China, Hefei, China

{jyxia, ysguan, xurk}@mail.ustc.edu.cn, zwxiong@ustc.edu.cn

## Abstract

Existing diffusion models excel at generating diverse content, but remain confined to fixed image shapes and lack the ability to flexibly control spatial attributes such as viewpoint, field-of-view (FOV), and resolution. To fill this gap, we propose Arbitrary-Shaped Image Generation (ASIG), the first generative framework that enables precise spatial attribute control while supporting high-quality synthesis across diverse image shapes (*e.g.*, perspective, panoramic, and fisheye). ASIG introduces two key innovations: (1) a mesh-based spherical latent diffusion to generate a complete scene representation, with seam enforcement denoising strategy to maintain semantic and spatial consistency across viewpoints; and (2) a spherical neural field to sample arbitrary regions from the scene representation with coordinate conditions, enabling distortion-free generation at flexible resolutions. To this end, ASIG enables precise control over spatial attributes within a unified framework, enabling high-quality generation across diverse image shapes. Experiments demonstrate clear improvements over prior methods specifically designed for individual shapes. Code is available at https://github.com/xjyjjy/ASIG.

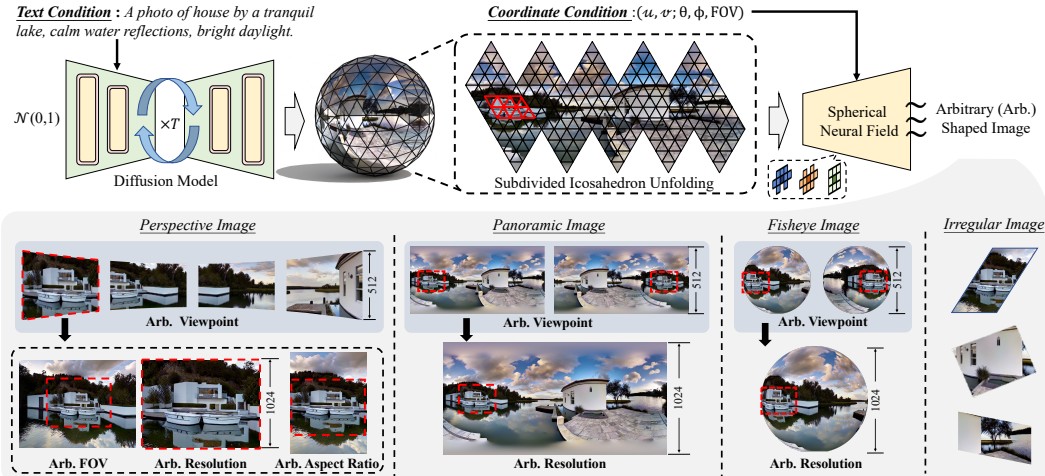

Figure 1: ASIG Overview. Our framework generates a complete spherical scene representation via a mesh-based spherical latent diffusion and enables distortion-free sampling through a spherical neural field. The proposed framework provides flexible control over FOV, viewpoint, and resolution, supporting high-quality generation across diverse image shapes, such as perspective, panoramic, and fisheye views. Generation at arbitrary aspect ratios naturally emerges from the joint control of FOV and resolution. As extended cases, irregular images can also be generated by jointly controlling FOV and resolution.

## 1 Introduction

Diffusion-based generative models have recently achieved remarkable success, powering a wide range of applications in artistic creation, design, and beyond (Ruiz et al., 2023; Brooks et al., 2023;

---
*Corresponding Author

Blattmann et al., 2023; Guan et al., 2025; Gao et al., 2024). However, a critical challenge lies in enabling precise and explicit control over the spatial attributes of the generated content. While Stable Diffusion (SD) (Rombach et al., 2022) allows text-conditioned image generation and SDXL (Podell et al., 2023) enables high-resolution generation, perspective image generation methods (Dhariwal & Nichol, 2021; Saharia et al., 2022; Peebles & Xie, 2023; Ramesh et al., 2022) still lack explicit control over viewpoint, field-of-view (FOV), and resolution. Recent arbitrary-resolution extensions (Chen et al., 2024; Kim & Kim, 2024) enable flexible output resolutions but retain unchanged FOV and viewpoint, producing merely upsampled detail rather than new scene content, which limits their practical effectiveness.

In parallel, panoramic generation methods (Tang et al., 2023; Zhang et al., 2024; Sun et al., 2025; Ni et al., 2025) can produce full-view imagery, but they remain tied to fixed-resolution panoramas at a single central viewpoint. New perspectives must be obtained through projection, which inevitably introduces distortion and fidelity loss. Thus, existing paradigms remain specialized to either perspective or panoramic generation and lack a unified framework that jointly controls viewpoint, FOV, and resolution while ensuring high-quality generation across different image shapes.

To fill this gap, we present ASIG, the first framework that unifies controllable generation of viewpoints, FOVs, and resolutions, while supporting diverse image shapes (*e.g.*, perspective, panoramic, and fisheye). ASIG introduces two key innovations: (1) a mesh-based spherical latent diffusion to generate a complete scene representation built on a multi-level subdivided icosahedron, with seam enforcement denoising strategy to maintain semantic and spatial consistency across viewpoints; and (2) a spherical neural field that incorporates feature extraction modules aligned with spherical topology, allowing arbitrary regions of the scene representation to be sampled under coordinate conditions, thereby enabling distortion-free generation at flexible resolutions.

To this end, ASIG establishs a unified framework for flexible, high-fidelity, and geometry-aware image generation. Beyond extending perspective generation with arbitrary FOVs and resolutions, ASIG naturally generalizes to panoramic and fisheye images, surpassing the generation quality of prior specialized generative methods.

Our main contributions are summarized as follows:

• We propose ASIG, the first unified diffusion-based framework that enables explicit control over viewpoint, FOV, and resolution, supporting diverse image shapes generation.
• We propose a mesh-based spherical latent diffusion that generates complete scene spherical representation with seam enforcement denoising to maintain semantic and spatial consistency across viewpoints.
• We introduce a spherical neural field that enables coordinate-conditioned, distortion-free sampling for arbitrary regions at flexible resolutions.
• Extensive experiments show that ASIG enables precise control over spatial attributes, generalizes across diverse image shapes, and surpasses specialized methods significantly.

## 2 RELATED WORK

**Perspective Image Generation** Perspective image generation with diffusion models has been extensively explored. Dhariwal & Nichol (2021) first demonstrated that diffusion models could surpass GANs in producing high-fidelity images, establishing diffusion as a powerful paradigm for image synthesis. Stable Diffusion (SD) and SDXL (Rombach et al., 2022; Podell et al., 2023) further advanced this direction by operating in latent space, incorporating text conditioning via cross-attention, and scaling effectively to high resolutions. However, they provide only weak spatial control. Image-GEN (Brooks et al., 2023) extends diffusion with instruction-driven editing capabilities, enabling controllable modifications but remaining confined to fixed image grids. Diffusion Transformers (DiT) (Peebles & Xie, 2023) replace convolutional UNets with pure Transformer backbones, achieving strong scalability and quality in perspective synthesis, yet they are still tied to fixed-resolution training and lack explicit viewpoint or FOV control. Overall, these methods implicitly capture correlations among resolution, content, FOV, and viewpoint from training data, without any explicit design for spatial attribute control, which fundamentally limits their ability to offer precise control.

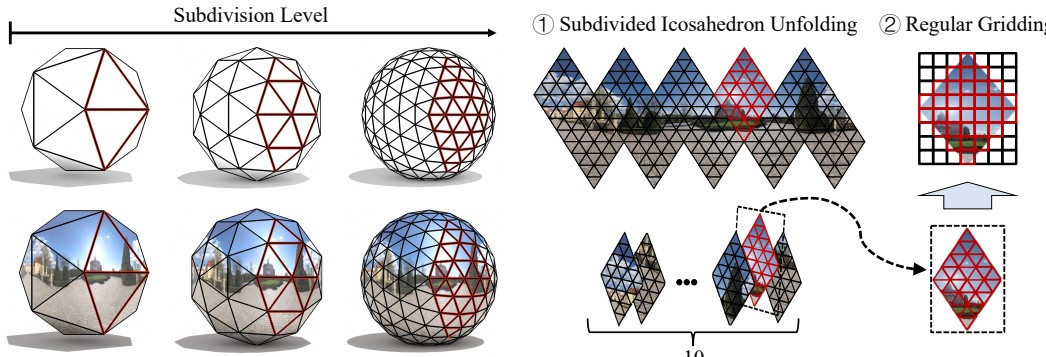

Figure 2: Illustration of L-subdivided icosahedron. Starting from a regular icosahedron (12 vertices, 20 triangular faces, 10 diamond-shaped faces), each subdivision quadrisects every face and reprojects the new vertices onto the sphere. This structure provides a distortion-free representation of the sphere, while the hierarchical subdivision naturally aligns with the multi-scale architecture of diffusion UNet, making it well-suited for generative modeling.

**Panoramic Image Generation**  Recent panoramic image generation methods mostly learning in the equirectangular projection (ERP) domain, but the inherent non-uniform distortions in ERP hinder accurate generative modeling. Stitching-based methods (Tang et al., 2023; Tuli et al., 2025) assemble perspective views into panoramas but suffer from limited FOV and cross-view inconsistencies. ERP-based diffusion methods (Zhang et al., 2024; Ni et al., 2025) directly train on panoramas, while cube- or sphere-aware variants (Kalischek et al., 2025; Sun et al., 2025) attempt to improve continuity via overlapping faces or spherical convolutions. Despite these advances, these methods remain tied to fixed rectangular grids, producing panoramas of fixed resolution from a single central viewpoint. Novel perspectives require projection, causing distortion and blur, while the non-uniform sampling of ERP further undermines semantic and spatial consistency.

**Neural Fields for Image Generation**  Neural fields represent signals with coordinate-based networks and have shown strong results in low-level vision tasks (Chen et al., 2021; Xu et al., 2025; Yao et al., 2024) and image generation. Prior methods (Chen et al., 2021; Lee et al., 2022; Yoon et al., 2022) adapt neural fields for arbitrary-scale super-resolution or spherical image representation. In image generation, INFD (Chen et al., 2024) and Kim & Kim (2024) further propose to generate perspective images at arbitrary scales, but remain limited to fixed aspect ratios, viewpoints, and FOVs. In contrast, we couple a spherical neural field with a mesh-based spherical latent diffusion, extending neural fields beyond arbitrary resolution to support arbitrary viewpoints and FOVs, while generalizing image generation from perspective to panoramic, fisheye images.

## 3  METHOD

**Overview**  ASIG integrates mesh-based spherical latent diffusion with a spherical neural field to achieve precise control over spatial attributes. During spherical latent diffusion, the UNet takes a text prompt together with an initial Gaussian noise tensor and progressively denoises it into the spherical latent, represented in the patch-based format shown in Fig. 2. As shown in Fig. 3, Seam-Aware Padding (SAP) enforces semantic consistency across neighboring patches, ensuring a complete spatial representation. SAP is likewise applied in the pherical neural field, where spherical residual blocks extract multi-scale features from the VAE decoder. Given coordinate conditions based on resolution, viewpoint, and FOV, a convolutional latent sampler maps the features to RGB values at the specified spatial attributes.

### 3.1  MESH-BASED SPHERICAL LATENT DIFFUSION

**Mesh-based Spherical Latent**  We introduce a mesh-based spherical latent representation built on an $L-$subdivided icosahedron. This representation aligns with the multi-scale hierarchy of the UNet in diffusion and the VAE model. As illustrated in Fig. 2, at subdivision level $L = 0$, each pair of adjacent triangular faces is merged into a diamond-shaped patch, resulting in 10 non-overlapping

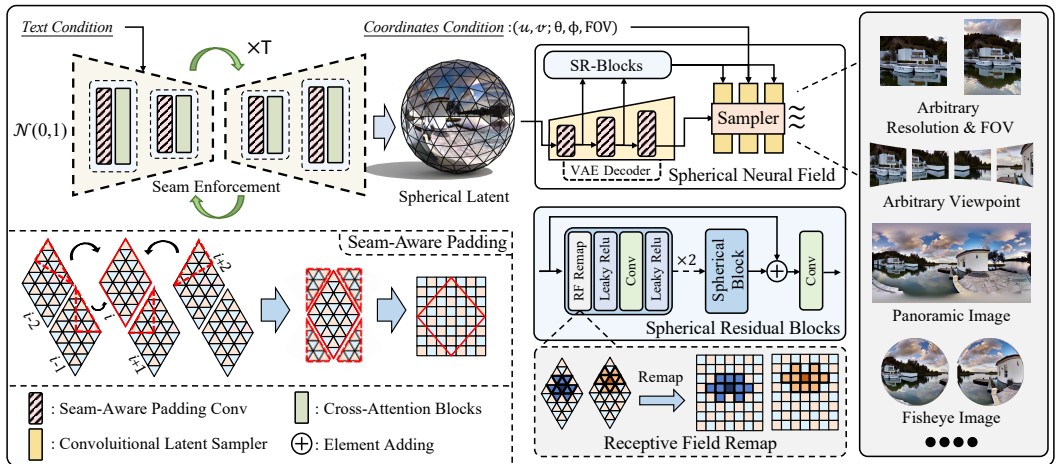

Figure 3: Illustration of the ASIG generation pipeline. Starting from Gaussian noise, a mesh-based spherical denoising UNet with Seam-Aware Padding performs $T$ denoising steps to produce a complete scene spherical latent aligned with the text condition. The latent is decoded by a VAE decoder to produce multi-scale features, which are processed by spherical residual (SR) blocks for hierarchical refinement. Finally, a coordinate-conditioned convolutional latent sampler generates high-quality images of diverse shapes.

patches that cover the entire sphere. For a general subdivision level $L$, each original face is further divided into $4^L$ triangular faces. Within each patch, we unfold the diamond into a rectangular grid and map every triangular face to a unique pixel by sampling its barycenter, thus establishing a one-to-one correspondence:

$$\mathcal{F}_L^p \longleftrightarrow \mathbf{M}_p \in \mathbb{R}^{H_L \times W_L \times d}, \quad p = 1, \dots, 10, \tag{1}$$

where $\mathcal{F}_L^p$ is the set of triangular faces in patch $p$, and $\mathbf{M}_p$ denotes its rectangular feature map. The spatial resolution of each patch grows with the subdivision level $L$:

$$(H_L, W_L) = (2^L, 2^L), \tag{2}$$

so that $L = 5$ yields $64 \times 64$ feature map and $L = 6$ yields $128 \times 128$ feature maps. This construction preserves spherical adjacency, maintaining feature integrity while enabling standard 2D convolutions on unfolded patches. More importantly, it naturally aligns with the multi-scale UNet of diffusion and VAE models, with subdivision levels corresponding to different feature resolutions.

**Seam-Aware Padding** However, each patch corresponds to a diamond-shaped region on the sphere. When unfolded into a rectangular grid, only the diamond area is valid while the remaining cells are zero-initialized. This leads to two issues: (*i*) convolutional kernels cannot access information across patch boundaries during inference, causing contextual discontinuities; and (*ii*) the zero-padded regions introduce artificial seams between adjacent patches.

$$\mathcal{N}(p_i) = \{\, p_{i-2},\ p_{i-1},\ p_{i+1},\ p_{i+2}\,\}, \tag{3}$$

with indices taken modulo 10 to respect the cyclic ordering of patches. As shown in Fig. 3, we explicitly annotate one example set of patch indices to illustrate how neighbors are arranged on the unfolded layout, and how they are used for Seam-Aware Padding. The padded feature map $\tilde{\mathbf{M}}_{p_i}(u,v)$ is then defined as

$$\tilde{\mathbf{M}}_{p_i}(u,v) = \begin{cases} \mathbf{M}_{p_i}(u,v), & (u,v) \in \text{valid region of } p_i, \\ \Pi_{p_i \leftarrow p_j}\big(\mathbf{M}_{p_j}(u',v')\big), & (u,v) \in \text{padded region, } p_j \in \mathcal{N}(p_i), \end{cases} \tag{4}$$

where $\Pi_{p_i \leftarrow p_j}$ denotes the geometric remapping along the mesh adjacency between $p_i$ and its neighbor $p_j$. In practice, we exploit the connectivity graph of the subdivided icosahedron to determine adjacency relations and boundary correspondences.

By applying Seam-Aware Padding to all convolutional layers in both the diffusion UNet and the VAE decoder, convolutional kernels perceive continuous features across patch boundaries while respecting spherical geometry. This strategy effectively removes artificial seams, ensures consistent semantics across the entire sphere, and allows the receptive field to extend smoothly over the mesh.

**Seam Enforcement Denoising**   To address discontinuities from independently processing unfolded spherical patches, we apply SAP throughout denoising. At each timestep, the latent $\mathbf{z}_t \sim \mathcal{N}(0, I)$ is augmented by copying boundary features from mesh-adjacent neighbors $\mathbf{x}_t = \mathrm{SeamPad}(\mathbf{z}_t; L)$, allowing convolutions to perceive continuous spherical context. The padded latent is passed to the UNet with timestep $t$ to predict noise, and the scheduler updates $\mathbf{z}_{t-1} = \mathrm{Step}(\mathbf{z}_t, \hat{\epsilon}_t, t)$. This process repeats until $t = 0$, yielding the seam-consistent spherical latent $\mathbf{z}_0$. Notably, $\mathrm{SeamPad}(\cdot; L)$ modifies only boundary cells while keeping interiors intact, ensuring both efficiency and semantic continuity.

## 3.2   SPHERICAL NEURAL FIELD

While the Mesh-based Spherical Latent Diffusion yields a consistent spherical latent $\mathcal{Z}_0$, direct decoding causes distortions and lacks explicit spatial control. To address this, we propose the Spherical Neural Field (SNF), which samples specified regions from $\mathcal{Z}_0$ under coordinate conditions. SNF respects spherical topology and consists of two parts: spherical residual blocks, which remap receptive fields to the mesh-based structure for multi-scale feature refinement, and a convolutional latent sampler for coordinate-based decoding.

**Spherical Residual Blocks**   Given the spherical latent $\mathcal{Z}_0$, we extract multi-scale decoder features $D^{(\ell)}(\mathcal{Z}_0)$. Each feature map is refined by Spherical Residual Blocks, where receptive fields are remapped to the icosahedron topology to reduce spherical distortions. $D^{(\ell)}(\mathcal{Z}_0)$ refined with spherical residual blocks ($\mathrm{SphRes}(\cdot)$), then upsampled to a common resolution ($\mathrm{Upsample}(\cdot)$), and finally merged along the channel dimension ($\mathrm{Concat}(\cdot)$) to form the multi-scale representation. Formally,

$$\mathbf{F} = \mathrm{Concat}_{\ell \in \{6,7,8,9\}}\Big( \mathrm{Upsample}\big(\mathrm{SphRes}^{(\ell)}(D^{(\ell)}(\mathcal{Z}_0))\big) \Big). \tag{5}$$

**Convolutional Latent Sampler**   After obtaining the multi-scale spherical features $\mathbf{F}$ and the VAE-decoded RGB image $D(\mathcal{Z}_0)$, we concatenate them along the channel dimension and apply coordinate-based sampling followed by a lightweight convolutional sampler to produce the final RGB output. Given a target viewpoint $(\theta, \phi)$ and FOV, we define a sampling region $\Omega(\theta, \phi, \mathrm{FOV})$ on the sphere. The projection function $\pi(u, v; \theta, \phi, \mathrm{FOV})$ maps each output pixel $(u, v)$ to its corresponding coordinate on the spherical latent, based on the chosen projection type and the sampling region $\Omega$. Using these coordinates, the rendered image is given by:

$$I(u, v) = f_\theta\Big(\mathrm{Sample}\big([\mathbf{F},\, D(\mathcal{Z}_0)],\, \pi(u, v; \theta, \phi, \mathrm{FOV})\big)\Big), \qquad (u, v) \in \mathcal{G}. \tag{6}$$

Here, $\mathrm{Sample}(\cdot)$ performs interpolation on the concatenated feature map at the projected coordinates, and $f_\theta$ is a lightweight network that maps the sampled features to RGB values. This formulation enables flexible image generation under arbitrary resolution, viewpoint, FOV, and image types.

## 3.3   TRAINING STRATEGY

**Stage 1: Spherical Neural Field Construction**   We freeze the VAE encoder and jointly train the spherical neural field in an end-to-end manner. Following Sec. 3.1, panoramas are converted into mesh-based patch, and ground-truth RGB images at arbitrary resolutions are sampled for supervision. The model is optimized with a weighted combination of pixel-wise loss $\mathcal{L}_1$, perceptual loss $\mathcal{L}_{\mathrm{LPIPS}}$, and adversarial loss $\mathcal{L}_{\mathrm{GAN}}$, balanced by coefficients $\lambda_1$, $\lambda_p$, and $\lambda_g$, respectively:

$$\mathcal{L} = \lambda_1\, \mathcal{L}_1 + \lambda_p\, \mathcal{L}_{\mathrm{LPIPS}} + \lambda_g\, \mathcal{L}_{\mathrm{GAN}}. \tag{7}$$

**Stage 2: Mesh-based Spherical Latent Diffusion Finetuning**   In the second stage, we train the UNet from Sec. 3.1 to model the reverse diffusion process, with the VAE encoder kept frozen. Each panorama is encoded into the mesh-based latent, perturbed with Gaussian noise, and the UNet is trained to predict the noise. To enforce cross-patch consistency, both the noisy latent $\mathbf{z}_t$ and ground-truth noise $\epsilon$ are augmented by the Seam-Aware Padding operator $\mathrm{SeamPad}(\cdot, L)$, yielding the training objective:

$$\mathcal{L}_{\mathrm{diff}} = \mathbb{E}_{t, \mathbf{z}_0, \epsilon}\Big[ \big\|\mathrm{SeamPad}(\epsilon, L) - F_\theta(\mathrm{SeamPad}(\mathbf{z}_t, L), t, y)\big\|_2^2 \Big], \tag{8}$$

where $F_\theta$ is the UNet denoiser and $y$ the text condition. This seam enforcement aligns training with inference and prevents boundary artifacts, ensuring semantic continuity across the sphere.

Table 1: Quantitative comparison with state-of-the-art methods on perspective, panoramic, and fisheye images. **Bold** and underline indicate the best and second-best results, respectively. KID* values are reported in units of $10^{-2}$.

| Metric | Perspective | | | | | | | Panorama | | | | Fisheye | | |
|---|---|---|---|---|---|---|---|---|---|---|---|---|---|---|
| | FID↓ | KID*↓ | Clip-FID↓ | NIQE↓ | PIQE↓ | MUSIQ↑ | BRIS.↓ | FID↓ | FAED↓ | KID*↓ | Clip-FID↓ | FID↓ | KID*↓ | Clip-FID↓ |
| INFD | 43.24 | 1.65 | 12.52 | 8.95 | 86.38 | 31.64 | 61.87 | 86.73 | 7.63 | 6.94 | 15.39 | 46.39 | 3.27 | 10.33 |
| Kim *et al.* | 40.77 | 1.87 | 11.86 | 8.46 | 88.61 | 30.98 | 64.98 | 83.88 | 9.87 | 7.77 | 17.96 | 47.86 | 3.47 | 10.56 |
| SD Pano. | 36.96 | 1.56 | 9.59 | 6.53 | 67.82 | 46.92 | 48.28 | 111.51 | 51.06 | 9.50 | 20.42 | 48.01 | 3.54 | 10.45 |
| SDXL Pano. | 35.97 | 1.55 | 11.00 | 5.57 | 65.39 | 53.16 | 41.36 | 82.13 | 7.53 | 4.90 | 16.50 | 45.13 | 3.03 | 10.04 |
| MVDiffusion | 23.17 | 0.93 | 5.84 | 5.92 | 69.29 | 47.53 | 44.33 | 82.24 | 54.74 | 6.71 | 9.91 | 31.21 | 2.40 | 5.26 |
| PanFusion | 21.33 | 0.99 | 5.44 | 8.04 | 86.73 | 32.86 | 60.42 | 37.09 | 1.63 | 1.59 | 3.64 | 17.16 | 0.87 | 2.78 |
| LayerPano3D | 53.22 | 3.09 | 13.16 | 7.68 | 82.12 | 39.81 | 54.44 | 66.16 | 5.02 | 3.86 | 12.47 | 48.34 | 3.12 | 10.79 |
| PAR | 37.16 | 1.92 | 6.14 | 9.38 | 89.91 | 28.45 | 65.52 | 35.50 | 1.53 | 1.29 | 6.30 | 28.56 | 2.04 | 4.14 |
| SMGD | 23.11 | 1.06 | 6.02 | 8.53 | 87.31 | 31.81 | 61.15 | 28.92 | 1.58 | 1.02 | 3.10 | 17.81 | 1.00 | 2.76 |
| Ours | 14.68 | 0.59 | 3.58 | 4.18 | 45.06 | 67.03 | 27.16 | 25.49 | 1.47 | 0.97 | 2.94 | 10.29 | 0.51 | 1.99 |

## 4 EXPERIMENT

### 4.1 TRAINING AND INFERENCE SETUP

The UNet and VAE weights are initialized from SDXL (Podell et al., 2023). Training proceeds in two stages. First, we freeze the VAE encoder and train the spherical neural field end-to-end for 100k iterations with AdamW (Loshchilov & Hutter, 2017) (batch size 4, learning rate $1.8\times10^{-5}$). Second, we train the UNet (Ronneberger et al., 2015) for another 100k iterations with mixed precision on 8×A100 GPUs (batch size 80), using a cosine annealing learning rate schedule between $3 \times 10^{-5}$ and $1 \times 10^{-7}$. Classifier-free guidance (Ho & Salimans, 2022) with 10% text dropout is applied. At inference, we use $\epsilon$-prediction with DDIM (Song et al., 2020) sampling for 50 steps.

### 4.2 DATASETS

Following previous methods (Tang et al., 2023; Zhang et al., 2024; Sun et al., 2025), we use the Matterport3D dataset (Chang et al., 2017), which contains 10,800 panoramas. Among them, 2,000 are reserved for testing, and the remaining are used for training. All text prompt are generated using BLIP-2 (Li et al., 2023a). During training, each panorama is preprocessed into mesh patches and converted into 10 patches for training.

### 4.3 EVALUATION

**Metrics** We evaluate our method across multiple image shapes. For perspective images, we report FID (Heusel et al., 2017), CLIP-FID (Kynkäänniemi et al., 2022), KID (Bińkowski et al., 2018), and NR-IQA metrics (NIQE (Mittal et al., 2012), PIQE (Venkatanath et al., 2015), MUSIQ (Ke et al., 2021), BRISQUE (Mittal et al., 2011)). For panoramas and fisheye images, we report FID, CLIP-FID, and KID, with FAED (Oh et al., 2022). Together, these metrics evaluate realism, perceptual quality, and geometric consistency across image shapes.

**Baseline Methods** We benchmark ASIG against a diverse set of state-of-the-art image generation approaches. The comparison includes panorama generation models MVDiffusion (Tang et al., 2023), PanFusion (Zhang et al., 2024), PAR (Wang et al., 2025), LayerPano3D (Yang et al., 2025) and SMGD (Sun et al., 2025), as well as arbitrary-scale generation methods including INFD (Chen et al., 2024) and Kim & Kim (2024). All models are trained on the same dataset to ensure fairness. In addition, we include two baselines, SDXL-panorama (Podell et al., 2023) and SD-panorama (Rombach et al., 2022), obtained by finetuning Stable Diffusion models with LoRA (Hu et al., 2022) on panoramic images.

**Quantitative Comparison** As reported in Tab. 1, ASIG consistently achieves the best performance across all metrics and image shapes. In particular, it yields substantial gains in FID, KID, and CLIP-FID for both perspective and fisheye generation, while also surpassing prior methods on panoramas. These results demonstrate that ASIG's unified design, combining mesh-based spherical latents with a spherical neural field, not only ensures semantic continuity across views but also enables precise control over viewpoint, FOV, and resolution. Overall, ASIG establishes a versatile and effective framework that outperforms all baselines.

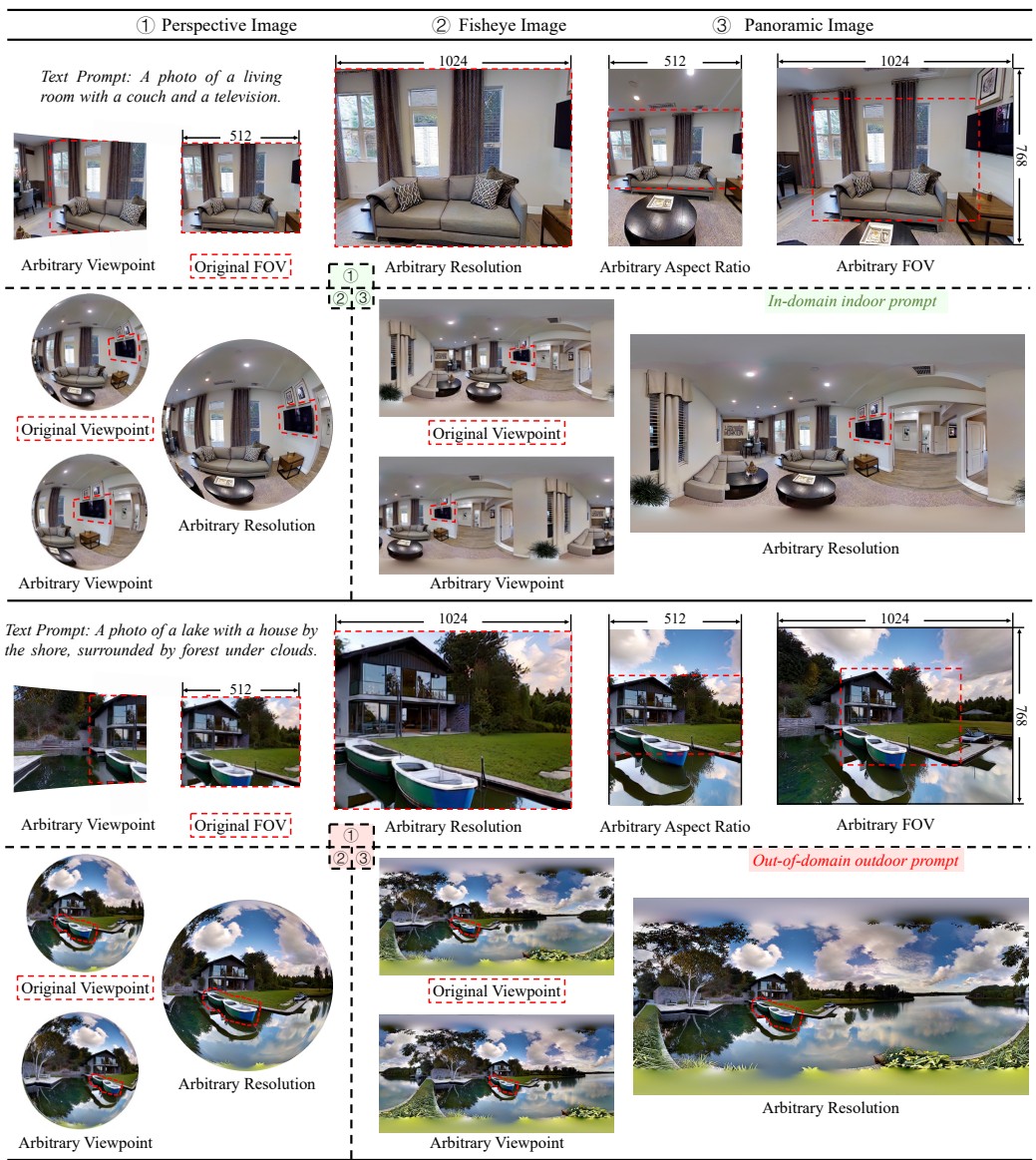

Figure 4: Qualitative results of ASIG on both in-domain (indoor) and out-of-domain (outdoor) scenes. ASIG produces high-quality perspective images with arbitrary viewpoint, FOV, resolution, and aspect ratio, as well as geometry-consistent fisheye and panoramic generations. It further demonstrates strong generalization to outdoor scenes beyond the training domain.

**Qualitative Comparison** In Fig. 4, we present qualitative results of ASIG across both in-domain and out-of-domain scenarios. For perspective images, ASIG supports arbitrary viewpoints, resolutions, aspect ratios, and FOVs, generating results that are not only geometrically consistent but also rich in fine details, free from the blurring or distortions. For fisheye and panoramic images, ASIG likewise enables arbitrary viewpoints and resolutions, while preserving seamless continuity across patch boundaries and delivering coherent global layouts. Benefiting from the mesh-based spherical latent, all projections are sampled from a unified representation, ensuring high fidelity, semantic consistency, and strong cross-view coherence across diverse image shapes.

In Fig. 5, we present visual comparisons with baselines on both perspective and panoramic image generation, including enlarged regions for closer inspection. ASIG consistently produces sharper and more realistic textures across all settings, highlighting its superiority over competing methods. Additional qualitative results, including extensive out-of-domain cases and further visual comparisons with baselines, are provided in the Appendix to more comprehensively demonstrate the robustness and generalization ability of our approach.

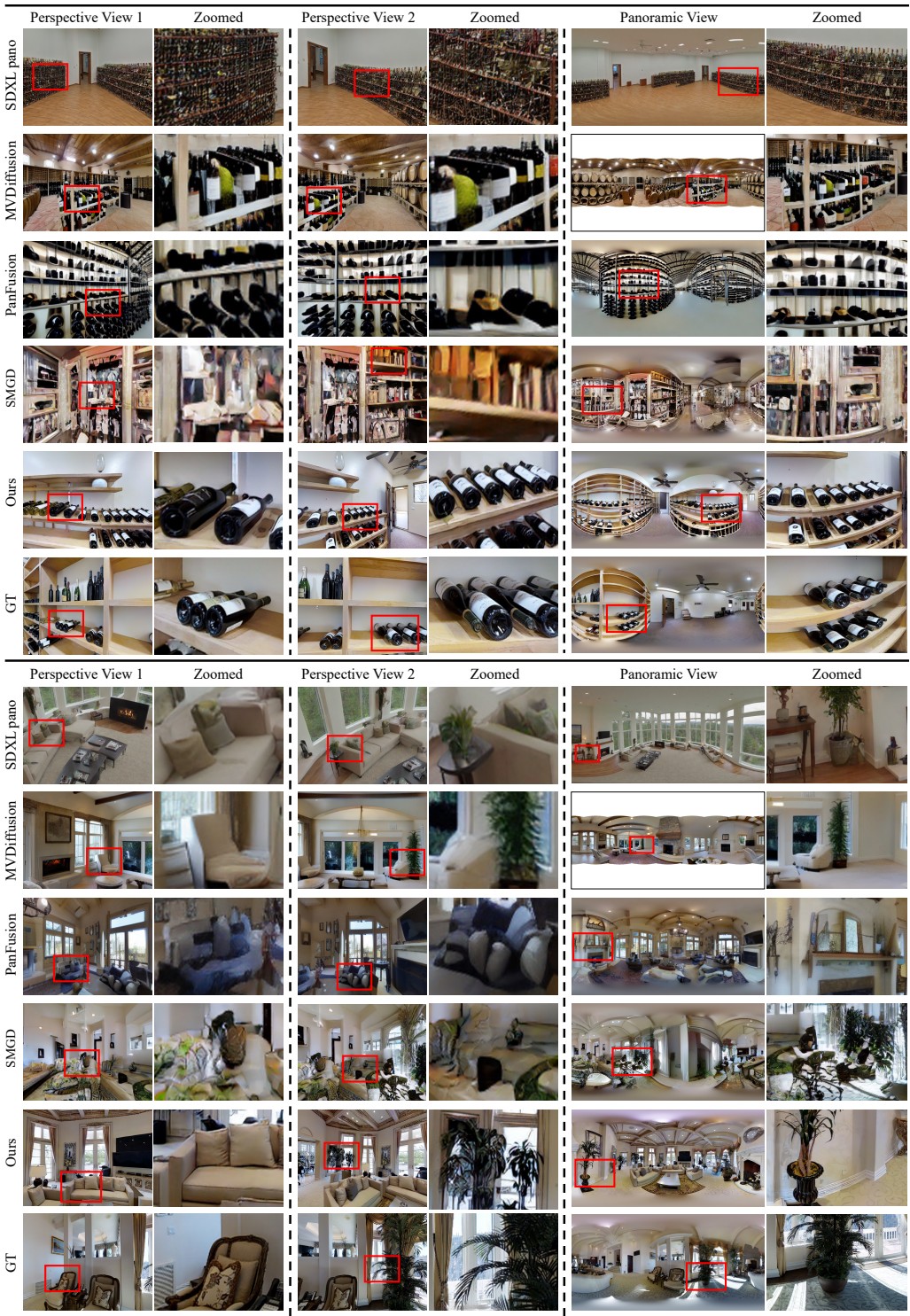

Figure 5: Qualitative comparison with baseline methods on perspective and panoramic image generation. The zoomed-in regions highlight that ASIG consistently produces sharper textures and more realistic details, while maintaining global consistency across viewpoints.

## 5 ABLATION

**Ablation on spherical neural field** To further validate the effectiveness of our spherical neural field, we enhance the baseline full-view generation model with LTEW (Lee et al., 2022). LTEW

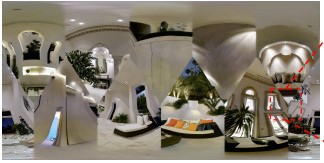 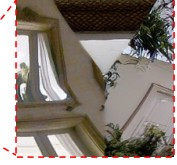 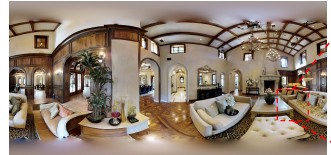 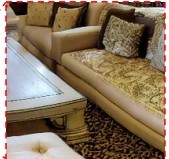

W/O SAP                                              W/ SAP

Figure 6: Visual ablation of Seam-Aware Padding (SAP) for panorama generation. Without SAP, seams and misalignment are clearly visible, whereas SAP effectively removes artifacts and ensures spatial consistency.

Table 2: Comparison on panorama quality across resolutions (pFID↓ / KID*↓). $^+$ denotes LTEW-enhanced variants (Lee et al., 2022). KID* values are reported in units of $10^{-2}$.

| Resolution | INFD | KIM | SDXL$^+$ | MVDiffusion$^+$ | PANfusion$^+$ | SMGD$^+$ | Ours |
|---|---|---|---|---|---|---|---|
| 1536 | 36.61 / 1.37 | 37.87 / 1.27 | 49.26 / 1.64 | 48.30 / 1.79 | 25.46 / 1.04 | 27.31 / 1.02 | **16.31 / 0.74** |
| 1024 | 42.63 / 1.49 | 41.86 / 1.54 | 52.79 / 1.73 | 57.99 / 2.03 | 27.66 / 1.07 | 29.46 / 1.13 | **18.76 / 0.86** |
| 512 | 56.36 / 1.78 | 52.38 / 1.88 | 59.84 / 1.97 | 64.26 / 2.76 | 29.24 / 1.15 | 31.32 / 1.24 | **20.49 / 0.93** |

Table 3: Ablation on spherical residual blocks. KID* values are reported in units of $10^{-2}$.

| SR Blocks | Reconstruction | | Generation | | |
|---|---|---|---|---|---|
| | PSNR↑ | LPIPS↓ | FID↓ | KID*↓ | CLIP-FID↓ |
| ✗ | 28.65 | 0.2205 | 27.93 | 0.99 | 3.03 |
| ✓ | 30.07 | 0.1680 | 25.49 | 0.97 | 2.94 |

Table 4: Ablation on Seam-Aware Padding in the spherical neural field. KID* values are reported in units of $10^{-2}$.

| SAP | Reconstruction | | Generation | | |
|---|---|---|---|---|---|
| | PSNR↑ | LPIPS↓ | FID↓ | KID*↓ | CLIP-FID↓ |
| ✗ | 29.17 | 0.1991 | 27.69 | 1.04 | 3.06 |
| ✓ | 30.07 | 0.1680 | 25.49 | 0.97 | 2.94 |

employs a super-resolution backbone to estimate local textures, which are then modulated by locally varying Jacobian matrices of coordinate transformations to predict Fourier responses (Tancik et al., 2020). This design improves the model's ability to handle geometric transformations such as homography and ERP-to-perspective projection. Across all tested resolutions, our spherical neural field achieves consistently lower patchFID (Chai et al., 2022) (pFID) compared to the LTEW-based baseline, demonstrating clear advantages in fidelity, scalability, and robustness for arbitrary-resolution and perspective image generation.

**Ablation on Spherical Residual Blocks**  We ablate the proposed spherical residual blocks on panoramic reconstruction and generation. Tab. 3 shows that removing spherical residual blocks lowers PSNR and raises LPIPS, indicating degraded fidelity, and also increases FID, KID, and Clip-FID, reflecting poorer realism. This confirms that spherical residual blocks are critical for geometry-aware feature extraction and maintaining consistency across resolutions.

**Ablation on Seam-aware Padding**  We evaluate the contribution of Seam-Aware Padding (SAP) by removing it separately in two components. Specifically, Tab. 4 reports results when removing SAP from the spherical neural field, while keeping it in the diffusion UNet. This leads to clear degradation in FID, KID, and CLIP-FID, confirming the necessity of SAP during spherical decoding. In addition, Fig. 6 shows qualitative comparisons when removing SAP from the diffusion UNet. Without SAP, visible seams and semantic misalignment appear at patch boundaries, whereas the full design produces seamless and consistent panoramas.

# 6    CONCLUSION

We introduced ASIG, the first diffusion-based framework that unifies controllable generation of viewpoint, FOV, and resolution across diverse image shapes. By combining mesh-based spherical latent diffusion for scene representation with a spherical neural field for distortion-free sampling, ASIG enables high-quality, geometry-consistent image synthesis across diverse shapes. Experiments demonstrate that ASIG achieves superior results on perspective, panoramic, and fisheye images, surpassing specialized baselines. Moreover, ASIG generalizes well to out-of-domain scenes, highlighting its robustness.

ACKNOWLEDGEMENTS

This work was supported in part by the National Natural Science Foundation of China under Grant 62131003, and the Fundamental Research Funds for the Central Universities under Grant WK2100000059.

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

APPENDIX

## A OUT-OF-DOMAIN GENERATION

We further generate 200 prompts for each Out-of-domain (OOD) scene type, covering a wide range of scene types, and compare our model against both SOTA panoramic generation methods (Zhang et al., 2024; Sun et al., 2025) and vanilla SDXL (Podell et al., 2023) on the CLIP-Score (CS) (Radford et al., 2021) metric. As shown in Tab. 5, our OOD CS metric remain close to vanilla SDXL and outperform SOTA panoramic baselines. We also assess generation quality (Wu et al., 2023; Mittal et al., 2012; 2011) across these scenes. As shown in Tab. 6, no performance gap under ID and OOD prompts condition; in all scenes, our method surpasses competing panoramic methods. Overall, these experiments confirm that our model retains strong open-domain generation capability while enabling arbitrary shape generation.

In Figs. 7 and 8, we present additional OOD visual results, including both panoramas and perspective views, which consistently demonstrate high quality and rich details. In Fig. 9, we provide additional OOD visual results across diverse scene types, further illustrating that our method maintains high visual quality even on OOD scenes.

## B MESH-BASED SPHERICAL RECONSTRUCTION PERFORMANCE AND COMPONENT OVERHEAD.

Our spherical representation has clear advantages over ERP and cubemap (Kalischek et al., 2025). ERP suffers from severe polar distortion, while cubemap introduces discontinuities and uneven sampling. In contrast, our mesh-based sphrical representation provides uniform sampling over the sphere, enabling the spherical neural field to behave consistently across viewpoints. As a result, we obtain more coherent geometry and better reconstruction quality. Tab. 7 shows higher PSNR and lower LPIPS (Zhang et al., 2018) than both baselines.

Moreover, the Seam-Aware Padding (SAP) tailored for spherical representations introduces no additional FLOPs and only minimal latency, since it consists solely of lightweight indexing and copying. As shown in the Tab. 8, our comparison between UNet with and without SAP indicates only a small runtime difference during inference (measured on A100). Importantly, once the spherical latent representation is generated, the spherical neural field (SNF) renders images at just 0.14s per image at 768×1024 resolution, while still supporting arbitrary viewpoints, FOVs, resolutions, and projection types. This demonstrates that the overall system remains efficient even with SAP.

## C FURTHER EXTENSIONS ON NEURAL FIELD TRAINING

We incorporated perspective images from the LSDIR (Li et al., 2023b) dataset into the training of our spherical neural field. Each perspective image is randomly projected onto our mesh-based

Table 5: Comparison of CLIP-Score (Radford et al., 2021) across various OOD scenes.

| OOD Scene | Nature | Fantasy | Weather | City | Complex | Avg. |
|---|---|---|---|---|---|---|
| PanFusion | 32.33 | 32.81 | 30.97 | 30.97 | 30.01 | 31.14 |
| SMGD | 23.77 | 26.30 | 22.22 | 24.83 | 28.46 | 25.12 |
| SDXL$_{Vanilla}$ | 32.98 | 33.43 | 32.51 | 31.39 | 31.05 | 32.27 |
| Ours | 32.84 | 33.23 | 31.23 | 31.02 | 30.71 | 31.81 |

Table 6: Comparison of no-reference metrics on ID and OOD scenes.

| Method | NIQE↓ | | BRISQUE↓ | | QA$_{quality}$↑ | | QA$_{aesthetic}$↑ | |
|---|---|---|---|---|---|---|---|---|
| | ID | OOD | ID | OOD | ID | OOD | ID | OOD |
| PanFusion | 4.74 | 4.92 | 38.13 | 38.06 | 3.99 | 3.87 | 3.51 | 3.71 |
| SMGD | 4.12 | 4.23 | 33.27 | 33.41 | 4.03 | 3.96 | 3.22 | 3.45 |
| Ours | **3.08** | **3.18** | **18.02** | **18.15** | **4.40** | **4.31** | **3.97** | **4.05** |

Table 7: Comparison of different panoramic representations for reconstruction quality.

| Method | PSNR↑ | LPIPS↓ |
|---|---|---|
| ERP | 29.07 | 0.1823 |
| Cubemap | 29.42 | 0.1788 |
| Ours | **30.07** | **0.1680** |

Table 8: Runtime comparison of UNet with and without SAP, and SNF rendering time.

| Component | Time (s) |
|---|---|
| UNet w/o SAP | 3.23 |
| UNet w SAP | 3.62 |
| SNF | 0.14 |

Table 9: Comparison of performance across both reconstruction and generation tasks.

| Train Data | Reconstruction | Generation |
|---|---|---|
| | PSNR↑ / LPIPS↓ | FID↓ / KID↓ |
| Matterport3D | 30.07 / 0.1680 | 25.49 / 0.97 |
| + LSDIR | **30.36 / 0.1655** | **25.22 / 0.94** |

spherical representation, and the spherical neural field is trained to reconstruct these images. As shown in Tab. 9, evaluations on both reconstruction and generation tasks show clear performance improvements compared to training only on Matterport3D(Chang et al., 2017).

## D    MORE VISUAL COMPARISON WITH BASELINE METHODS

In Fig. 10, we show further visual comparisons with baselines. ASIG consistently outperforms them on both perspective and panorama generation, delivering richer details, more realistic textures, and uniform image quality.

## E    ADDITIONAL VISUAL RESULTS ON SD3/DIT BACKBONES

In Fig. 11, we present additional visual results produced by SD3/DiT-based (Esser et al., 2024) models. These samples demonstrate excellent texture quality and color fidelity, further showing that our method is transferable across different diffusion backbones.

## F    THE USE OF LARGE LANGUAGE MODELS

We employed Qwen-3 to refine sentence clarity and check spelling. No large language model was employed for generating experimental results, figures, or core technical claims. All content was reviewed, verified, and approved by the human authors.

## G    ETHICS STATEMENT

We confirm that this work adheres to the ICLR Code of Ethics. It does not involve human subjects, sensitive data, or practices that raise ethical concerns.

## H    REPRODUCIBILITY STATEMENT

We detail the components of our method in Sec. 3 and the experimental settings in Sec. 4, and provide extensive visual results in Sec. 4 and the Appendix for verification. We will release the training and testing code together with the processed datasets upon publication to ensure full reproducibility.

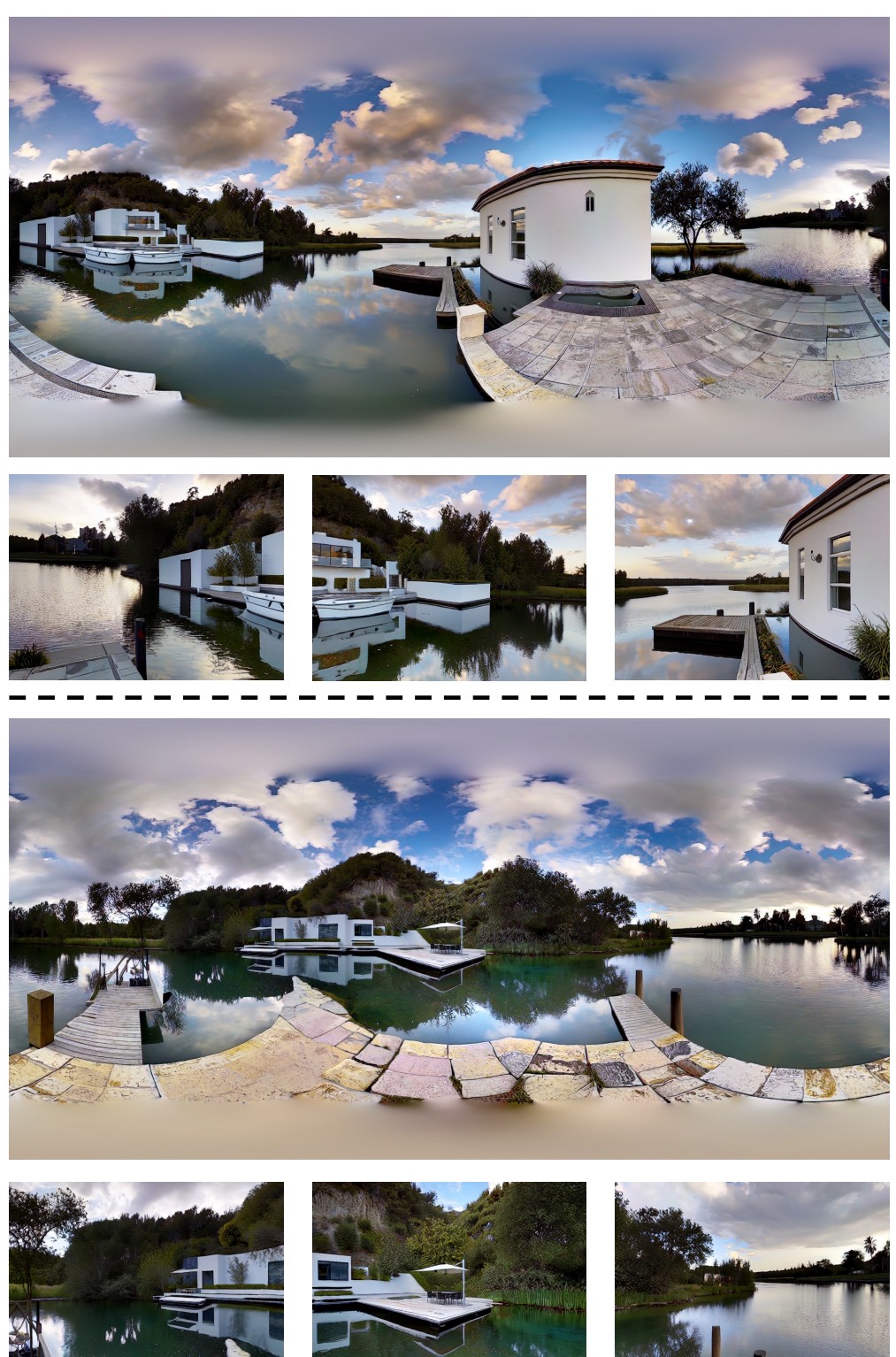

Figure 7: Out-of-domain examples (panoramas and perspective views) generated by ASIG with the text prompt: "A photo of a house by a tranquil lake, with calm water, bright daylight."

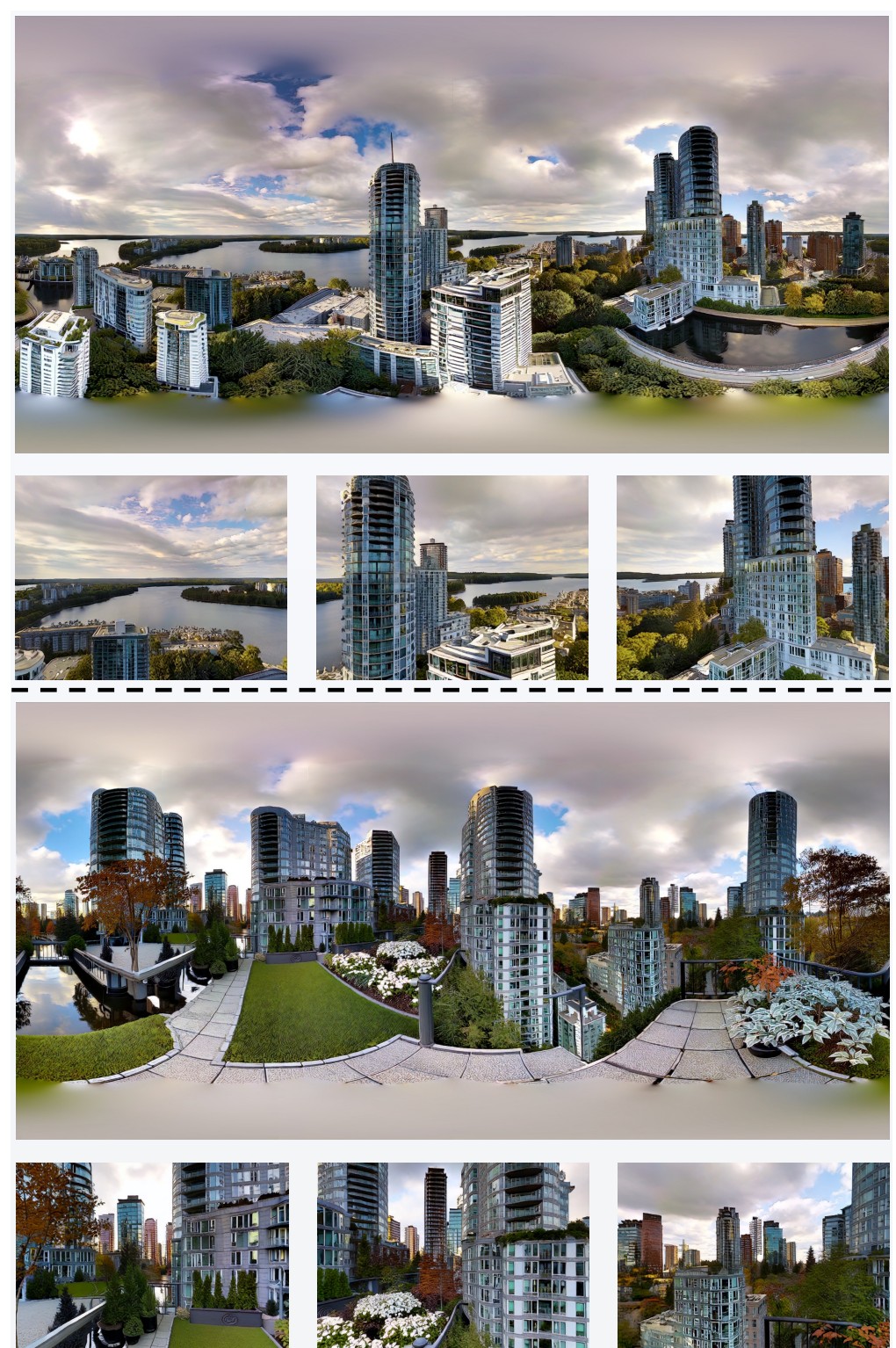

Figure 8: Out-of-domain examples (panoramas and perspective views) generated by ASIG with the text prompt: "A photo of forest and skyscrapers, with a lake under sunny white clouds."

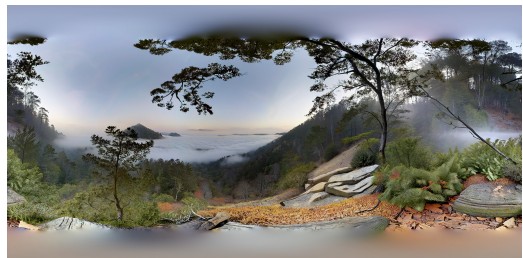

*Natute Prompt*: a photo of a panoramic mountain landscape at sunrise with misty valleys and tall trees.

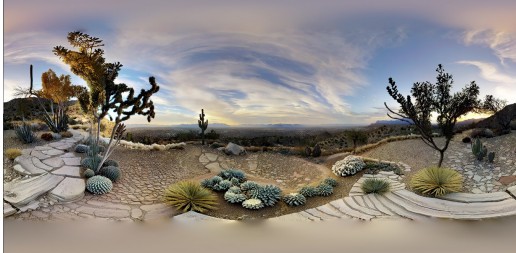

*Natute Prompt*: a photo of a desert panorama with sparse plants under a dramatic cloudy sky.

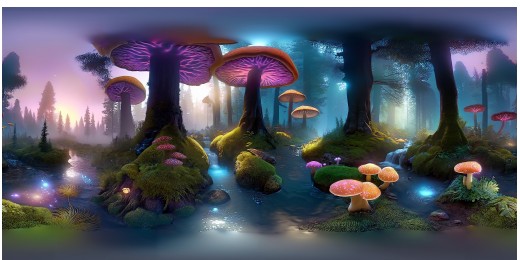

*Fantasy Prompt*: a photo of a fantasy forest panorama with glowing giant mushrooms and colorful lights.

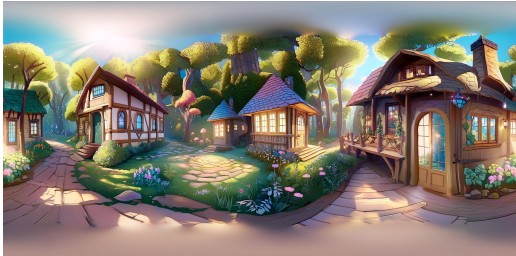

*Fantasy Prompt*: a photo of a fairytale village panorama with wooden cottages and warm sunlight.

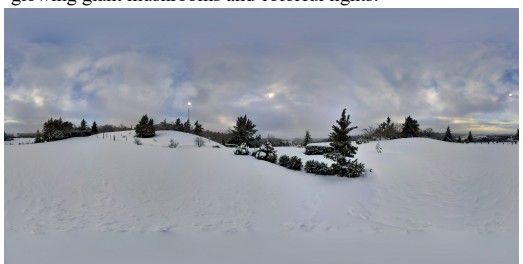

*Weather Prompt*: a photo of a snow-covered panoramic field under a calm overcast winter sky.

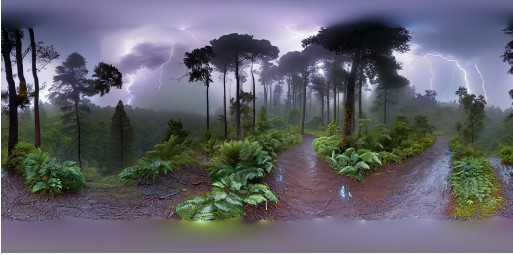

*Weather Prompt*: a photo of a forest panorama during a thunderstorm with bright lightning and wet foliage.

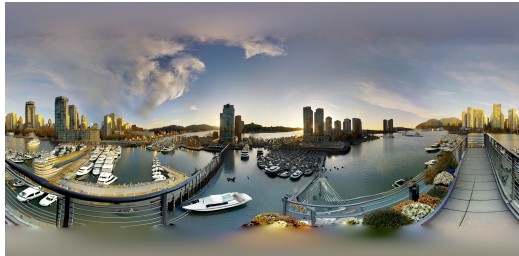

*City Prompt*: a photo of a harbor city panorama at sunset with yachts, bridges, and golden reflections.

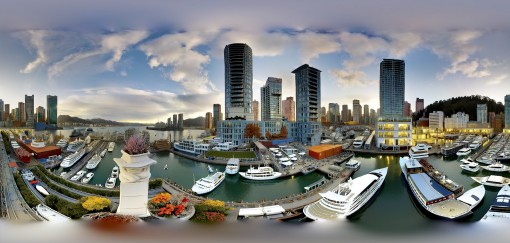

*City Prompt*: a photo of an urban waterfront panorama with skyscrapers rising above a busy marina.

Figure 9: Out-of-domain examples covering diverse outdoor scene types (nature, fantasy, weather, and city). Although these scenes are far from the training domain (indoor Matterport3D), our method produces high-quality and semantically coherent panoramic generations.

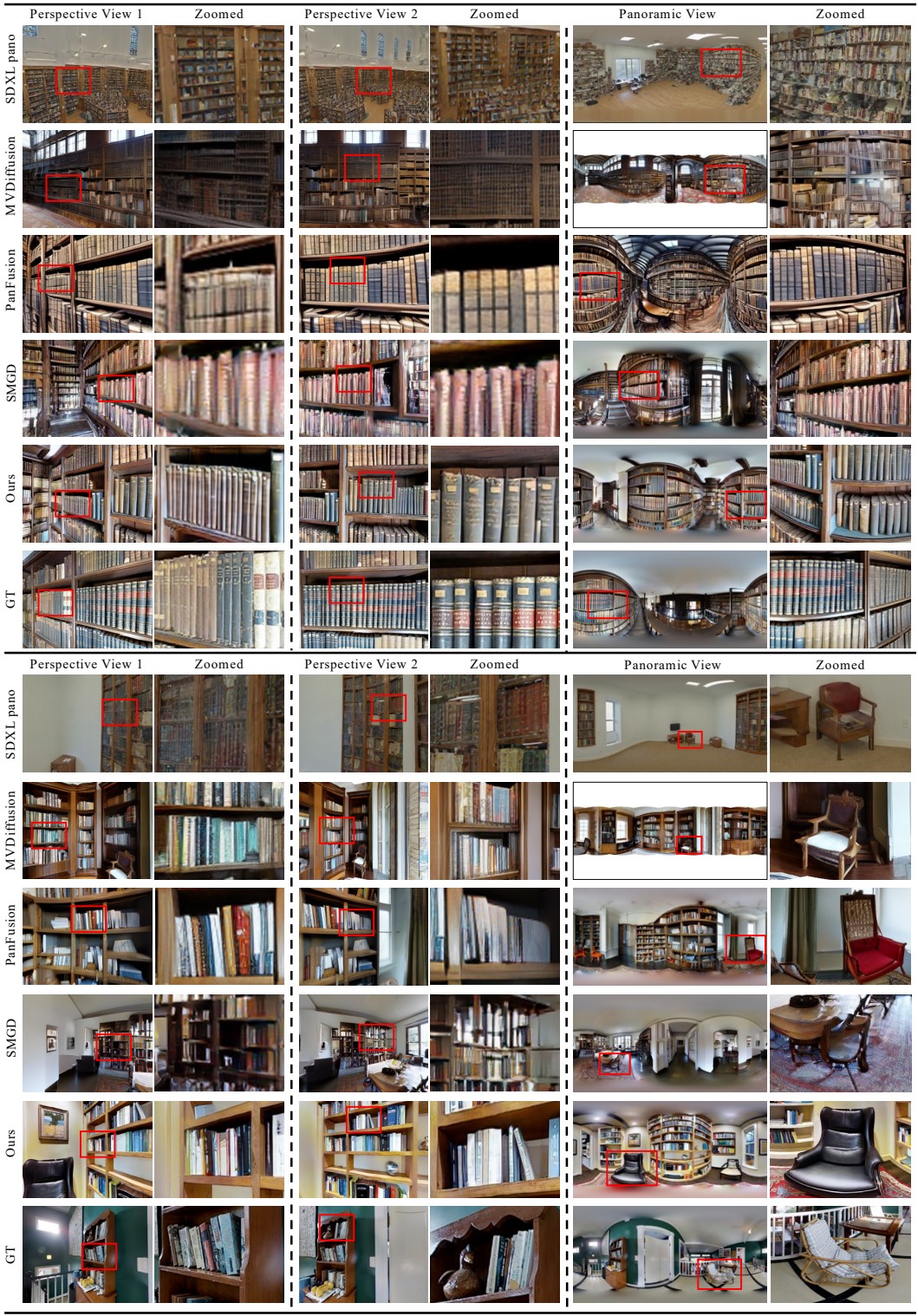

Figure 10: Qualitative comparison with baseline methods on perspective and panorama generation. ASIG produces richer details, more realistic textures, and more consistent image quality across views.

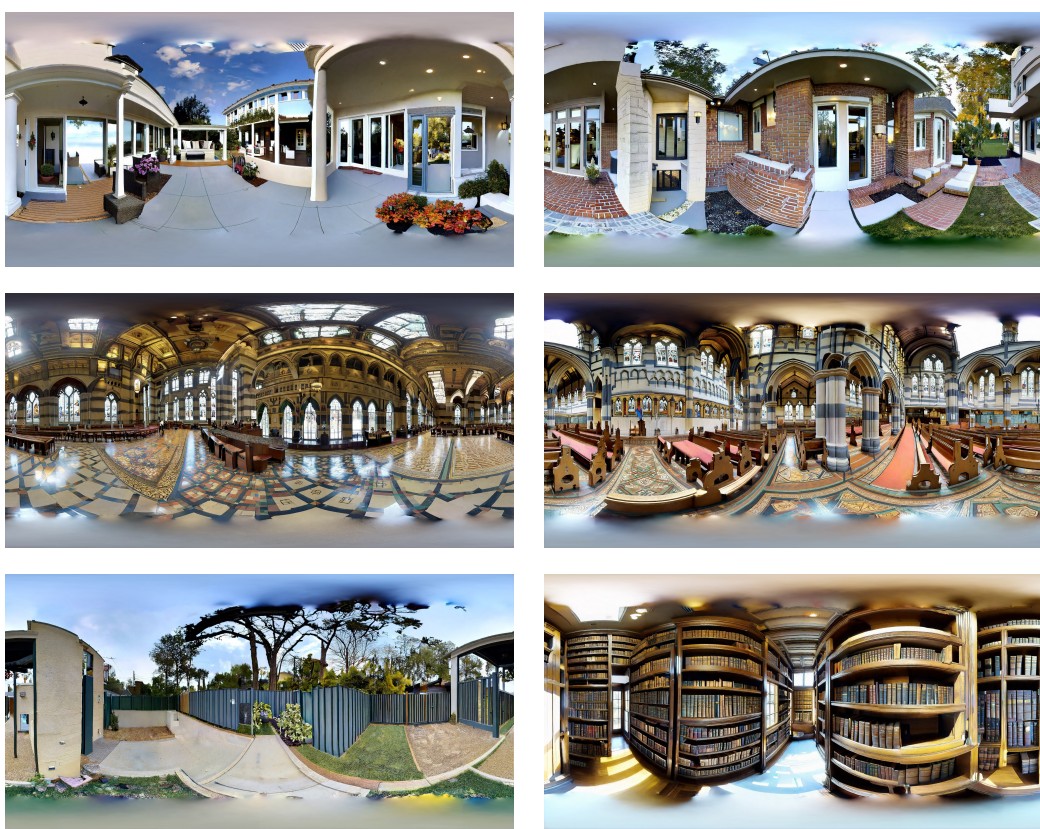

Figure 11: Visual results generated by the DiT-based model, showing high-quality textures, consistent colors, and overall strong visual fidelity.

