# OpenReview forum: "Arbitrary-Shaped Image Generation via Spherical Neural Field Diffusion"
_ICLR.cc/2026/Conference — ICLR 2026 Poster_

### Official Review · Reviewer_6hfe · 2025-10-27

**Soundness:** 3
**Presentation:** 3
**Contribution:** 3
**Rating:** 6
**Confidence:** 4

**Summary:**

This paper proposes Arbitrary-Shaped Image Generation (ASIG), whose core idea is to use the latent of an implicit field representation as the denoising target, enabling resolution- and field-of-view-independent image generation. The method introduces Mesh-based Spherical Latent, Seam-aware Padding, and Seam Enforcement Denoising for the design of the diffusion representation, model, and network architecture.

**Strengths:**

1. This paper proposes an interesting framework for resolution-independent image generation. Both the paper and the framework are easy to follow and understand.
2. Although using implicit representations to handle varying resolutions is an effective and commonly adopted approach, the representation, network, and model design proposed in this paper are well-motivated and meaningful.
3. The paper provides detailed and comprehensive quantitative experiments to demonstrate the effectiveness of the model, achieving state-of-the-art performance on multiple tasks, including panoramic, perspective, and fisheye image generation.

**Weaknesses:**

1. Considering that DiT-based architectures are currently more mainstream and have demonstrated superior performance, using a U-Net–style architecture might be somewhat outdated. The authors could further explore attention-based operators under their proposed representation to enrich the paper’s content and potentially improve model performance.
2. One major advantage of implicit representations, beyond enabling resolution-independent inference, is that during training they could also leverage diverse types of images (e.g., standard perspective images) to augment the dataset and improve model generalization. The paper currently lacks exploration in this direction, but I believe this could be a promising avenue for further performance improvement.
3. Although the operators designed for the spherical grid are reasonable, and the authors cleverly convert them into regular tensors, I question how practical these methods are for real-world implementation and deployment, especially since the authors have not released the model code.

**Questions:**

Please see the weaknesses.

Overall, my assessment of this paper is positive. I look forward to the authors’ responses to my concerns and remain open to the possibility of raising my score.

**Details Of Ethics Concerns:**

No.

---

> ### Author Response · Authors · 2025-11-21
> **Response to Reviewer 6hfe**
>
> We thank reviewer **6hfe** for the constructive and insightful feedback. Your comments are **extremely valuable** and have helped us further improve technical quality. We are glad to provide further responses if there still remain any concerns.
>
> ### **W1.Further Exploration on DiT**
>
> Following your recommendation, we extended our framework to the **SD3** DiT-based architecture. This adaptation is **non-trivial**. Specifically, for every attention block, we unpatchify its output to (B, C, H, W), apply Seam-Aware Padding, and then repatchify the result before feeding it into the next attention block. We retrained both the VAE and the DiT transformer weights to ensure compatibility with our spherical representation.
>
> Since the **main contribution** of our work is **not** the **diffusion backbone**, we have not fully optimized the DiT performance—certain training details such as learning rate, guidance scale, and the text dropout ratio for classifier-free guidance are still under exploration. Nevertheless, the current results, as reported in **Tab. 1***, already demonstrate that our spherical representation integrates smoothly with the DiT architecture and achieves performance comparable to the UNet-based version. In addition, it achieves competitive visual quality, including realistic textures and fine details, as illustrated in Fig. 11 of the supplementary material.
>
> We hope this addresses your concern and appreciate your valuable suggestion.
>
> |           | **FID↓** | **FAED↓** | **KID↓** | **Clip-FID↓** |
> | :-------: | :------: | :-------: | :------: | :-----------: |
> | Ours_SDXL |  25.49   |   1.47    |   0.97   |     2.94      |
> | Ours_SD3  |  29.27   |   1.53    |   1.03   |     2.83      |
>
> **Table 1\*:** Quantitative comparison of SDXL (UNet)–based and SD3 (DiT)–based methods on panoramic image generation.
>
> ### **W2.Further Extensions on Implicit Neural Representation Training**
>
> We agree that implicit neural representations can also leverage diverse image types (e.g., perspective images) to expand the training data and improve the model’s generalization ability. Your comment strongly inspired us to conduct additional experiments.
>
> Specifically, we incorporated perspective images from the LSDIR dataset into the training of our spherical neural field. Each perspective image is randomly projected onto our mesh-based spherical representation, and the spherical neural field is trained to reconstruct these images. As shown in **Tab. 2***, evaluations on both reconstruction and generation tasks show clear performance improvements compared to training only on Matterport3D.
>
> We will continue expanding this idea by integrating larger and more diverse datasets to further enhance generalization ability. We sincerely appreciate your **valuable** suggestion.
>
> |      Task      |     Metric     | only  Matterport3D | Matterport3D + LSDIR |
> | :------------: | :------------: | :----------------: | :------------------: |
> | Reconstruction | PSNR↑ / LPIPS↓ |   30.07 / 0.1680   |    30.36 / 0.1655    |
> |   Generation   |  FID↓ / KID↓   |    25.49 / 0.97    |     25.22 / 0.94     |
>
> **Table 2\*:** Comparison of performance across both reconstruction and generation tasks.
>
> ### **W3.Practicality and Open-Source Commitment**
>
> Regarding practicality, our mesh-based spherical representation offers several tangible advantages —uniform sampling, seamless topology, consistent receptive fields, and projection-agnostic rendering. These properties make the spherical neural field much easier to optimize and lead to higher-quality reconstruction and generation.
>
> In terms of efficiency, the framework is also highly practical. As shown in **Tab. 3***, the spherical latent is produced at nearly the same speed as vanilla SDXL, as the additional components Seam-aware Padding introduced for spherical-latent generation add only negligible overhead. Importantly, once the spherical latent representation is generated, the spherical neural field (SNF) renders images at just **0.14s per image** at 768×1024 resolution, while still supporting arbitrary viewpoints, FOVs, resolutions, and projection types.
>
> Moreover, we have added additional generated results across diverse scenes in **Fig. 9** of the supplementary material, which further demonstrates the practicality and versatility of our framework.
>
> Finally, we confirm that all **code** and **checkpoints** will be open-sourced upon acceptance, ensuring full reproducibility and accessibility for the community.
>
> |          | UNet w/o SAP | UNet w SAP | SNF  |
> | :------: | :----------: | :--------: | :--: |
> | Time (s) |     3.23     |    3.62    | 0.14 |
>
> **Table 3\*:** Runtime comparison of UNet inference with and without SAP, and the spherical neural field (SNF) rendering time.

---

> > ### Comment · Reviewer_6hfe · 2025-11-22
> >
> > Thanks for the authors' efforts; most of my concerns have been resolved. I encourage and look forward to the authors continuing to explore the DiT structure and the extension of the model using a wide range of regular image data.
> >
> > I am keeping my score for now, but later in the discussion phase, after weighing the quality and scores of all the papers in my batch, I may raise my score.
> >
> > Once again, thank you to the authors for the detailed and thorough supplementary experiments and responses; I deeply feel your dedication.

---

> > > ### Author Response · Authors · 2025-11-22
> > >
> > > We are glad that our responses adequately addressed your concerns. Thank you for the positive recommendation.

---

### Official Review · Reviewer_L4Gf · 2025-10-30

**Soundness:** 3
**Presentation:** 3
**Contribution:** 4
**Rating:** 6
**Confidence:** 2

**Summary:**

This paper introduces ASIG, a diffusion-based framework for arbitrary-shaped image generation, which unifies viewpoint, field-of-view (FOV), and resolution control within a single model. ASIG combines two main components: (1) Mesh-based spherical latent diffusion representing the scene over a subdivided icosahedron mesh with a seam-aware padding (SAP) mechanism to ensure spatial consistency. (2) Spherical neural field (SNF) – decoding the spherical latent representation into images with arbitrary projections (perspective, panoramic, fisheye) via coordinate-conditioned sampling.

**Strengths:**

1. Solid Technical Design. The authors designed a new representation for the spherical diffusion model and achieved arbitrary viewpoint and resolution generation through improved VAE.

2. The unification of perspective, panorama, and fisheye generation in one framework is interesting.

**Weaknesses:**

1. Seam-aware padding is a clever design, but it seems to significantly increase the complexity of the model during computation.

2. Although SAP and SR blocks are ablated, the contribution of the mesh subdivision is not studied. One thing I'm curious about is what advantages this representation method has compared to many commonly used Cube representation methods? It seems that many of the claimed advantages are due to the addition of control signals in the VAE decoding process, rather than this representation method itself.

3. Lack of comparison with some of the latest methods, such as LayerPano3d[1] and PAR[2] :

[1] LayerPano3D: Layered 3D Panorama for Hyper-Immersive Scene Generation
[2] Conditional panoramic image generation via masked autoregressive modeling

**Questions:**

See weakness.

---

> ### Author Response · Authors · 2025-11-21
> **Response to Reviewer L4Gf**
>
> We thank the reviewer **L4Gf** for the constructive and insightful feedback. Below we respond to each point in detail and will incorporate the suggested improvements in the revised version. We are glad to provide further responses if there still remain any concerns.
>
> ### **W1. The Complexity of Seam-Aware Padding (SAP)**
>
> SAP introduces **no additional FLOPs** and only **a little latency**, since it consists solely of lightweight indexing and copying operations. As shown in the **Tab. 1*** , our comparison between UNet with and without SAP indicates only a **small runtime difference** during inference  (measured on A100). Importantly, once the spherical latent representation is generated, the spherical neural field (SNF) renders images at just **0.14s** per image at 768×1024 resolution, while still supporting arbitrary viewpoints, FOVs, resolutions, and projection types. This demonstrates that the overall system remains highly efficient even with SAP.
>
> |          | UNet w/o SAP | UNet w SAP | SNF  |
> | :------: | :----------: | :--------: | :--: |
> | Time (s) |     3.23     |    3.62    | 0.14 |
>
> **Table. 1\*:** Runtime comparison of UNet inference with and without SAP, and the spherical neural field (SNF) rendering time.
>
> ### **W2. The Advantage of Mesh-based Spherical Representation**
>
> Our spherical representation offers clear advantages over other methods. ERP introduces severe polar distortion, and cubemap suffers from discontinuities and uneven sampling across faces. In contrast, our mesh-based spherical representation provides geometrically uniform sampling over the whole sphere. This uniformity allows the model to use a spherical neural field that behaves consistently across all viewpoints, instead of forcing the network to adapt to the large distortion variations seen in ERP and cubemap.  As a result, our method produces more coherent geometry and higher reconstruction quality. In our panoramic reconstruction experiments, as shown in **Tab. 2***, it achieves higher PSNR and lower LPIPS than both ERP and cubemap, demonstrating more faithful preservation of spherical structure.
>
> |         | PSNR  | LPIPS  |
> | :-----: | :---: | :----: |
> |   ERP   | 29.07 | 0.1823 |
> | Cubemap | 29.42 | 0.1788 |
> |  Ours   | 30.07 | 0.1680 |
>
> **Table. 2\*:** Comparison of different panoramic representations for reconstruction quality.
>
> ### **W3. More Baselines**
>
> For baseline selection, we chose SMGD because it is the strongest existing method according to prior publications. Following your suggestion, we additionally evaluated two other representative baselines PAR and LayerPano3D. Their performance is consistently weaker than SMGD. We now report metrics for panoramic, perspective, and fisheye images for all baselines, and have updated the revised manuscript accordingly.
>
> |  Panorama   |   FID↓    |  FAED↓   |  KID*↓   | Clip-FID↓ |
> | :---------: | :-------: | :------: | :------: | :-------: |
> | LayerPano3D |   66.16   |   5.02   |   3.86   |   12.47   |
> |     PAR     |   35.50   |   1.53   |   1.29   |   6.30    |
> |    SMGD     |   28.92   |   1.58   |   1.02   |   3.10    |
> |    Ours     | **25.49** | **1.47** | **0.97** | **2.94**  |
>
> **Table. 3\*:** Quantitative comparison with baseline methods on panoramic image.
>
> | Perspective |   FID↓    |  KID*↓   | Clip-FID↓ |  NIQE↓   |   PIQE↓   |  MUSIQ↑   | BRISQUE.↓ |
> | :---------: | :-------: | :------: | :-------: | :------: | :-------: | :-------: | :-------: |
> | LayerPano3D |   53.22   |   3.09   |   13.16   |   7.68   |   82.12   |   39.81   |   54.44   |
> |     PAR     |   37.16   |   1.92   |   6.14    |   9.38   |   89.91   |   28.45   |   65.52   |
> |    SMGD     |   23.11   |   1.06   |   6.02    |   8.53   |   87.31   |   31.81   |   61.15   |
> |    Ours     | **14.68** | **0.59** | **3.58**  | **4.18** | **45.06** | **67.03** | **27.16** |
>
> **Table. 4\*:** Quantitative comparison with baseline methods on perspective image.
>
> |   Fisheye   |   FID↓    |  KID*↓   | Clip-FID↓ |
> | :---------: | :-------: | :------: | :-------: |
> | LayerPano3D |   48.34   |   3.12   |   10.79   |
> |     PAR     |   28.56   |   2.04   |   4.14    |
> |    SMGD     |   17.81   |   1.00   |   2.76    |
> |    Ours     | **10.29** | **0.51** | **1.99**  |
>
> **Table. 5\*:** Quantitative comparison with baseline methods on fisheye image.

---

### Official Review · Reviewer_73SH · 2025-10-30

**Soundness:** 3
**Presentation:** 3
**Contribution:** 3
**Rating:** 6
**Confidence:** 3

**Summary:**

This paper proposes ASIG, a novel diffusion framework designed to overcome the limitations of existing models by enabling explicit control over viewpoint, FOV, and resolution in a unified system. The method first generates a complete spherical scene representation via a mesh-based latent diffusion, and then uses a spherical neural field to sample and render arbitrary-shaped images (e.g., perspective, panoramic). The quantitative and qualitative results shows clear improvements over current methods.

**Strengths:**

1. ASIG enables explicit control over viewpoint, FOV, and resolution within a unified system. Both quantitative and qualitative results confirm that the method outperforms existing approaches.

2. The authors' methodology is reasonable. ASIG constructs a spherical latent space by linking UNet principles with an $L$-subdivided icosahedron, and the proposed seam-aware padding (SAP) and spherical neural field components are effective in achieving excellent results.

**Weaknesses:**

1. Some ablation results are confusing.

	a. The reported results for ASIG in Table 2 are inconsistent with the corresponding results in Table 1.

	b. Furthermore, the ASIG results presented in Table 2 also differ from those in Table 3 and Table 4.

2. The Sampler component is underspecified.

	a. The Sampler appears to integrate two distinct information sources: a 'remapped mesh-structure' and 'information processed by SAP'. It is unclear how these sources are fused or prioritized. Is there conflict between them and how they merge together?

	b. The process for inputting the 'sampling region' information is not detailed. Please elaborate on how this region is defined and fed into the sampler.

3. The manuscript's clarity needs significant improvement, as several sections are difficult to understand.

	a. The explanation for Equation 3 + line 204 is insufficient. The authors should provide a more detailed derivation or justification.

	b. The description accompanying Equation 5 (line 248), particularly the phrase "F guided by..." (or "F guided by $\pi$"?), is ambiguous. It is not clear what this operation entails or how the guidance mechanism works.

4. Some important details are missing.

	a. Whether the sampling strategy requires an additional stitching mechanism or post-processing step to ensure seamless transitions between different sampled patches?

	b. In SAP component, transforming rectangular inputs into squares is unclear. How is this achieved (e.g., resizing?), and does this step introduce unwanted distortion?

	c. What is the meaning of "out-of-distribution" (OOD) (Figure 14&15).

	d. How to apply different projection functions π(·)?

**Questions:**

1. Why the 'Receptive Field Remap' is needed, given that the VAE encoder already processes features using 'SAP'?

2. Where do the weights of the VAE encoder come from? Are the other models that need to be trained trained from scratch?

---

> ### Author Response · Authors · 2025-11-21
> **Response to Reviewer 73SH**
>
> We thank reviewer **73SH** for the helpful comments. Some misunderstandings and suggestions are addressed below. We are glad to provide further responses if there still remain any concerns.
>
> ### **W1. Ablation Results**
>
> **a and b.** Note that Tab. 2 of the manuscript reports pFID, where panoramas are cropped into patches for evaluation. In contrast, Tabs. 1, 3, and 4 of the main manuscript report full-image FID. Because the evaluation metrics differ, the resulting numbers naturally differ as well.
>
> ### **W2. Detail of Sampler Component**
>
> **a.** The “information processed by SAP” refers to the RGB image obtained after the VAE decoder. In Eq. 5, **F** has shape (B, 12, H, W) because the feature $D^{(\ell)}(Z _0)$ is refined by the Spherical Residual Blocks, upsampled to the same resolution, and concatenated. This feature map is then concatenated with the RGB image and fed into the sampler. We have corrected Eq. 6 in the revised manuscript. Thank you for pointing it out.
>
> **b.** The “sampling region”, as described in the manuscript, is defined the sampling region as $\ \Omega(\theta, \phi, \mathrm{FOV})$, where the center viewpoint is given by longitude $\theta$ and latitude $\phi$, and the FOV determines the span of longitudes and latitudes covered. Once $\Omega$ is fixed, we construct a (2, H, W) coordinate map, where each 2D coordinate indexes into the spherical latent in the unfolded 2D feature space. This coordinate map is then used by the sampler to fetch features and render the final RGB image.
>
> ### **W3. Manuscript Clarity**
>
> **a.** The expression $ \mathcal{N}(p_i)= \lbrace p_{i-2},p_{i-1},p_{i+1},p_{i+2} \rbrace$  specifies the patch indices from which information is used to fill the diamond-shaped empty regions during seam-aware padding. To avoid ambiguity, we have now explicitly annotated the example patch indices in Fig. 3 (SAP illustration) and **added a description at line 203 in the revised manuscript** explaining how $\mathcal{N}(p_i)$ relates to the numbering of the unfolded patches.
>
> **b.** For Eq. 5, the $\pi(u,v;\theta,\phi,\mathrm{FOV})$ creates the sampling coordinates determined jointly by the chosen sampling region and the projection type, as described in our response to **W2.b**. In practice, these coordinates specify where each pixel in the target image should query the spherical latent. The expression **F** guided by $\pi$ means that we directly sample the feature tensor **F** at sampling coordinates. We have added a clear explanation of this notation in the revised manuscript.
>
> ### **W4. Missing Details**
>
> **a.** Our sampling strategy does not require any additional stitching or post-processing. Thanks to the diamond-grid topology and explicit cross-face contextual conditioning, the spherical latent is inherently seamless, and all sampled views remain smooth without extra stitching.  All our visual results consistently support this point.
>
> **b.** As reviewer **L4Gf** noted, Seam-Aware Padding (SAP) is a clever design. As illustrated in Fig. 3, each diamond-shaped spherical face is composed of pixel rows of lengths $1,3,5,\dots,2n-1,2n-1,\dots,5,3,1$. This layout naturally unfolds into the red-outlined square region on the right without any resizing or distortion. We use light-orange and -blue blocks to explicitly illustrate the one-to-one correspondence.
>
> **c.** We would like to clarify that in our original submission, the meaning of OOD was already explained in Fig. 4 and its caption. We also note that there is no “Figure 14&15” in the paper. Out-of-domain (OOD) refers to scenes outside the training domain, i.e., outdoor scenes, since the model is trained on indoor Matterport3D images.
>
> **d.** All projection functions in our method are based on standard spherical projections, such as sphere-to-perspective, equirectangular, and fisheye projections. As described in W3.b, a projection function $\pi$ creates sampling coordinates map each pixel (u,v) in the target image to the sphere latent. Our implementation follows established projection formulas, and the code for all projection types will be released upon acceptance.
>
> ### **Q1. The Importance of "Receptive Field Remap"**
>
> SAP preserves spherical topology and enables cross-face contextual conditioning, allowing the diffusion model to build a coherent spherical latent. However, rendering a 2D image still requires the spherical neural field to adapt its receptive field to the spherical structure, which is precisely what the Receptive Field Remap (RFR) ensures. As shown in Tab. 3 of the manuscript, removing the spherical residual blocks (which contain RFR) noticeably degrades both reconstruction and generation, indicating that SAP and RFR address different stages and are both necessary.
>
> ### **Q2. Training Details**
>
> As stated in Section 4.1 of the original submission, all model weights are initialized from SDXL. Only the VAE encoder is frozen, while all other components are fully trained after SDXL initialization.

---

### Official Review · Reviewer_mcAc · 2025-10-31

**Soundness:** 3
**Presentation:** 3
**Contribution:** 2
**Rating:** 6
**Confidence:** 4

**Summary:**

This paper presents a method for generating images with arbitrary shapes (different aspect ratios, etc.) and projections (e.g. panoramic, fisheye) form a single model. It defines a 2D latent diffusion over a sphere around the camera, and rasterises from this to create the final image. Standard U-net architectures are applied on the sphere by discretising it as a modified icosahedron, subdividing triangles, and noting that pairs of triangles form diamonds having a grid topology. Beyond this the method is a standard latent diffusion. The model is initialised from SDXL, and fine-tuned on the Matterport3D dataset.

**Strengths:**

The proposed approach is novel; latent diffusion over a discretised sphere is an elegant approach to learning a generative model over 2D visual content that is invariant to the camera projection type and parameters. The specific approach to mapping the sphere to grid topology (so pretrained UNets) can be used is also interesting.

Results are shown using a single model to generate diverse images – perspective at diverse aspect ratios, panoramic, and fisheye. Quality is comparable for all image types, with no visible degradation e.g. in highly deformed regions.

Compared with several baselines for panoramic/large/multiview image generation, the proposed method achieves stronger results for three projection types across seven metrics; FID in particular is significantly lower than any other methods.

There is a fairly detailed ablation study, measuring the benefit of several of the introduced components, including the spherical residual block, and use of a spherical representation instead of direct deformation of planar generations.

The model is shown to retain some of the open-domain characteristics of the original SDXL model it is fine-tuned from (despite that fine-tuning only using the single-domain Matterport3D dataset) – in particular, it is still possible to generate out-of-distribution images given text conditioning, even under fisheye/etc projections.

**Weaknesses:**

The overall technical innovation is rather small. The tricks required to apply the diffusion UNet on the sphere are not particularly noteworthy (e.g. there is significant discussion of what simply amounts to circular convolution). Beyond this, the model is essentially a standard latent diffusion model.

It is unclear how far the ability to generate arbitrary scenes (as opposed to those in-distribution for Matterport3D) is retained. While one qualitative example is shown, in addition there should be a quantitative comparison (e.g. using CLIP score) on a broad range of prompts, and this compared against the vanilla SDXL.

The claim of "arbitrary resolution" is rather strong. The method is trained using a fixed discretisation of the sphere, and while this can be sampled at arbitrarily high spatial frequencies, the fineness of synthesised details is bounded by the discretisation (and the training data) – unlike truly "infinite resolution" models like Generative Powers of Ten. The present work is similar to training a very high resolution 2D diffusion then cropping out an ROI of the desired resolution (though of course the spherical representation allows other lens types etc.).

**Questions:**

Please address the points mentioned under "Weaknesses" above – in particular, measure more thoroughly how well open-domain generation works with the fine-tuned model.

---

> ### Author Response · Authors · 2025-11-21
> **Response to Reviewer mcAc (Part 1/2)**
>
> We thank the reviewer **mcAc** for the constructive and insightful feedback. Below we respond to each point in detail and will incorporate the suggested improvements in the revised version. We are glad to provide further responses if there still remain any concerns.
>
> ### **W1. Technical Innovation**
>
> While we find in the strengths that you commented " the proposed approach is novel ", the technical innovation of this work has also been well recognized by **all other** reviewers  (**73SH**/**L4Gf**/and **6hfe** commented **“methodology is reasonable”/ “solid technical design”** / **“well-motivated and meaningful”).**
>
> Specifically, we are the first to perform diffusion directly in a mesh-based spherical latent space, and we further design a dedicated spherical neural field that enables high-quality generation of images with arbitrary shapes.
>
> It is worth mentioning that applying a diffusion UNet on a spherical latent is far from straightforward. We identify several key challenges in applying a diffusion UNet to a spherical latent and explore many geometric design options before finalizing the unfolding and projection strategy in Fig. 2.
>
>  The key challenges include:
>
> - Projection distortion. The projection process often introduces uneven distortion, as seen in common formats such as ERP and cubemap [1]. Through many tested projection layouts, we found that the diamond topology effectively reduces projection distortion, as its face arrangement distributes sampling more evenly.
> - Cross-face contextual conditioning. A spherical representation requires consistent contextual information across adjacent faces. The diamond-shaped corner connections enable adjacent faces to exchange contextual cues, providing effective conditioning for maintaining semantic consistency.
> - Projection information retention. The projection process often leads to information loss due to interpolation operations. Our projection format is specifically designed so that each mesh pixel maps one-to-one into the rectangular grid, effectively preventing information loss.
> - Subdivision level. The subdivision level needs to balance reconstruction detail and computational cost. We also tested multiple subdivision levels and selected the configuration that best balances representation quality and computational cost.
>
> Overall, our spherical representation achieves better PSNR and LPIPS than ERP and cubemap in panoramic reconstruction, further validating the soundness of our design.
>
> |         | PSNR↑ | LPIPS↓ |
> | :-----: | :---: | :----: |
> |   ERP   | 29.07 | 0.1823 |
> | Cubemap | 29.42 | 0.1788 |
> |  Ours   | 30.07 | 0.1680 |
>
> **Table 1\*:** Comparison of different panoramic representations for reconstruction quality.

---

> ### Author Response · Authors · 2025-11-21
> **Response to Reviewer mcAc (Part 2/2)**
>
> ### **W2. Open-Domain Generation Ability**
>
> We have already included many **out-of-domain** **(OOD)** examples in Figs. 4, 7, 8 of the manuscript, and their visual quality remains on par with **in-domain** **(ID)** results. To address your concern more thoroughly, we further generate 200 prompts for each OOD scene type, covering a wide range of scene types, and compare our model against both SOTA panoramic generation methods and vanilla SDXL on the **CLIP-Score (CS)** metric.  As shown in **Tab. 2***, our OOD CS scores remain close to vanilla SDXL and outperform all panoramic baselines.
>
> |  OOD Scene   | Nature | Fantasy | Weather | City  | Complex | Avg.  |
> | :----------: | :----: | :-----: | :-----: | :---: | :-----: | :---: |
> |  PanFusion   | 32.33  |  32.81  |  30.97  | 30.97 |  30.01  | 31.14 |
> |     SMGD     | 23.77  |  26.30  |  22.22  | 24.83 |  28.46  | 25.12 |
> | SDXL_Vanilla | 32.98  |  33.43  |  32.51  | 31.39 |  31.05  | 32.27 |
> |     Ours     | 32.84  |  33.23  |  31.23  | 31.02 |  30.71  | 31.81 |
>
> **Table 2\*:** Comparison of CLIP-Score across various OOD scenes.
>
> We also assess generation quality across these scenes. As shown in **Tab. 3*** , no performance gap under **ID** and **OOD** prompts condition; in all scenes, our method surpasses competing panoramic methods. We also add representative visual results in **Fig. 9** of the supplementary. Overall, these experiments confirm that our model retains strong open-domain generation capability while enabling arbitrary shape generation ability.
>
> |  Method   | **NIQE↓ (ID)** | **NIQE↓ (OOD)** | **BRISQUE↓ (ID)** | **BRISQUE↓ (OOD)** | **QA_quality↑ (ID)** | **QA_quality↑ (OOD)** | **QA_aesthetic↑ (ID)** | **QA_aesthetic↑ (OOD)** |
> | :-------: | :------------: | :-------------: | :---------------: | :----------------: | :------------------: | :-------------------: | :--------------------: | :---------------------: |
> | PanFusion |      4.74      |      4.92       |       38.13       |       38.06        |         3.99         |         3.87          |          3.51          |          3.71           |
> |   SMGD    |      4.12      |      4.23       |       33.27       |       33.41        |         4.03         |         3.96          |          3.22          |          3.45           |
> |   Ours    |    **3.08**    |    **3.18**     |     **18.02**     |     **18.15**      |       **4.40**       |       **4.31**        |        **3.97**        |        **4.05**         |
>
> **Table 3\*:** Comparison of no-reference metrics on ID and OOD scenes.
>
> ### **W3. Task Setting of Arbitrary Resolution**
>
> We would like to further clarify this point to avoid any misunderstanding. Our framework can render a specified ROI at arbitrary resolutions while maintaining consistent visual quality, rather than simply cropping the ROI from a higher-resolution image.  As shown in Tab. 2 of the manuscript, the generated image quality continues to improve as the resolution increases, and our method achieves SOTA performance across all resolutions.
>
> Moreover, the strong generalization of implicit neural representations to high-resolution image has been widely demonstrated in prior works (e.g., LIIF [2], INFD [3]). Following these works, our reconstruction stage is trained with images at a diverse range of resolutions.
>
> These results further confirm that our approach supports true resolution-agnostic generation. This capability enables us to generate images of arbitrary shapes while consistently preserving visual quality across different viewpoints, FOVs, and projection types.
>
> [1] Kalischek, Nikolai, et al. "Cubediff: Repurposing diffusion-based image models for panorama generation." *The Thirteenth International Conference on Learning Representations*. 2025.
>
> [2] Chen, Yinbo, Sifei Liu, and Xiaolong Wang. "Learning continuous image representation with local implicit image function." Proceedings of the IEEE/CVF conference on computer vision and pattern recognition. 2021.
>
> [3] Chen, Yinbo, et al. "Image neural field diffusion models." *Proceedings of the IEEE/CVF Conference on Computer Vision and Pattern Recognition*. 2024.

---

### Author Response · Authors · 2025-12-02
**Concluding Remarks**

Dear Area Chairs,

Thank you for your time and consideration of our submission. All reviewers assigned positive ratings to our submission.(**mcAc**/**73SH**/**L4Gf** and **6hfe** commented **"the proposed approach is novel"**/**"methodology is reasonable"**/**"solid technical design"**/**"well-motivated and meaningful"**.)

During the rebuttal period, we responded to **all reviewers’ concerns and questions**, including several additional analyses to **further highlight** our advantages, such as showing that our method preserves **sufficient open-domain generation capability** inherited from vanilla SDXL and demonstrating its applicability to a DiT backbone, among others. All relevant experiments have been incorporated into the revised manuscript.

Reviewer **6hfe** expressed **satisfaction** with our responses and indicated that they were considering **increasing their score**. Meanwhile, we have addressed all other reviewers’ questions, including those stemming from misunderstandings and open-ended suggestions, and anticipate that they will find our responses satisfactory.

We would appreciate it if you could take this context, including the **possibility of score increases**, into account in your evaluation.

Thank you!

---

### Meta-Review · Area_Chair_PRUD · 2026-01-03

**Summary:**

The paper presents a method to achieve text-to-image generation with explicit control over arbitrary shapes (resolution, aspect ratio etc.) and projections (panoramic, fisheye etc.) within a unified model. The model is built-upon a diffusion-based framework. The paper initially receives 4 boarderline accepts (i.e. rating 6), and the reviews from four reviewers suggest that 1) the proposed approach is novel and reasonable; 2) solid technical design; 3) achieves stronger results than prior methods; 4) fairly detailed ablation studies; 5) provides detailed and comprehensive quantitative experiments. Although there are certain concerns initially: 1) technical innovation is small; 2) performance regarding the OOD generation; 3) require further clarification on certain parts, etc., the AC think the rebuttals have addressed most of these concerns, by providing further clarifications and providing more experimental evaluations. Therefore, the AC is happy to recommend the acceptance of the paper.

**Reviewer Concerns:**

The AC think the authors have well addressed all the critical concerns of the reviewers. They provided detailed additional experimental results and further clarifications.

**Reviewer Scores:**

Based on the raised concerns, overall quality of the work, confidence of the reviewer as well as the rebuttal from the authors, the AC think: 1) mcAc might keep original rating; 2) 73SH might keep original rating; 3) L4Gf would keep original rating; 4) 6hfe would possibly increase his rating.

---

### Decision · Program_Chairs · 2026-01-26

Accept (Poster)